# A helping hand: Applying behavioural science and co-design methodology to improve hand hygiene compliance in the hospital setting

Calea-Jay Blair[1], Clare McCrudden[2], Alix Brazier[3], Sarah Huf [1], Alice Gregory[2], Fiona O'Driscoll [2], Tracey Galletly[4], Clare Leon-Villapalos[4], Helen Brown[3], Kristina Clay[4], Shona Maxwell[4], Raymond Anakwe[4], Kate Grailey [5]*

1 Institute of Global Health Innovation, Imperial College London, London, United Kingdom, 2 Helix Centre, Institute of Global Health Innovation, Imperial College London, London, United Kingdom, 3 Behavioural Insights Team, London, United Kingdom, 4 Imperial College Healthcare NHS Trust, London, United Kingdom, 5 Centre for Health Policy, Institute of Global Health Innovation, Imperial College London, London, United Kingdom

* k.grailey18@imperial.ac.uk

## Abstract

Compliance with hand hygiene is an effective way of reducing the incidence of healthcare acquired infections (HCAI). At one London National Health Service (NHS) Trust, improving hand hygiene compliance (HHC) was a patient safety priority in response to non-compliance and ongoing occurrences of HCAI. The objective of this study was to co-design a behavioural science informed intervention to improve HHC. To obtain a baseline level of HHC and understand associated behaviours, 18 hours of observation were undertaken on three inpatient wards. These focused upon Moment 1 and 5 of the World Health Organisation's moments for hand hygiene. The intervention was co-designed with clinical staff and took the form of "visual primes". Three different stickers designed to create a motivational "nudge" were placed at key points where HHC had been observed to fail. Following implementation, a further 18 hours of observation took place. A Chi-squared statistical analysis compared proportions of HHC pre- and post-intervention. Our intervention led to an 11% increase in HHC across the three study wards for both Moments ($X^2$ (1, N = 1,285) = 13.711, p = <0.001) in the six weeks following the intervention. The intervention had a more marked effect on Moment 1, (with an increase of 15%, $X^2$ (1, N = 667) = 17.091, p = <0.001 when compared to the change in compliance with Moment 5 (11%, $X^2$ (1, N = 652) = 7.449, p = 0.06). This study demonstrated that utilising behavioural science in the co-design and placement of visual motivational nudges can significantly improve compliance with hand hygiene practices. We highlight the benefit of co-design when designing interventions—both in terms of engagement with and efficacy of the intervention.

## Background

Healthcare Acquired Infections (HCAI) are infections that arise as a consequence of the transmission of microbes within the hospital setting, risking increased patient morbidity and

**Data Availability Statement:** All data underlying the findings in the study are available within the article.

**Funding:** The author(s) received no specific funding for this work.

**Competing interests:** The authors have declared that no competing interests exist.

mortality [1, 2]. Additionally, they are a financial burden on the healthcare system due to increased length of stay and intensity of treatment required [3]. There is growing evidence that if implemented properly, hand hygiene practices can reduce the risk of HCAI's [3].

Such practices include correct hand washing, using alcohol-based hand gels, the use of personal protective equipment (PPE) and maintaining a clean environment [4, 5]. The World Health Organisation (WHO) has created a patient safety programme called "SAVE LIVES: Clean Your Hands" [5], demonstrating the importance of correct HHC. The WHO's "5 Moments for Hand Hygiene" is a comprehensive framework designed to provide guidance to healthcare professionals in maintaining rigorous hand hygiene practices that prevent the spread of infections [4].

However, compliance with hand hygiene protocols can be lacking. Major reasons for non-compliance with hand hygiene include forgetfulness and sanitiser unavailability, with possible contributing factors being a lack of reminders and understaffing [6]. Additional factors influencing hand hygiene compliance (HHC) include variations in practice between healthcare workers of different disciplines, patient location, frequency and nature of audits, and differing practices required for different clinical scenarios [7]. Peer influence is also important–with perceived peer handwashing frequency influencing an individual's own behaviour [8].

The importance of correct hand hygiene to prevent disease transmission was globally stressed during the COVID-19 pandemic—both to the public and healthcare professionals [9]. Changes to practice as a result of COVID-19 led to new ways of working with respect to both personal protective equipment (PPE) and hand hygiene–with the use of gloves becoming universal [10, 11]. It has been noted post-pandemic that some confusion between the need for HHC when using gloves has been demonstrated–with some staff choosing to use gloves as an alternative to HHC [12]. This is despite unwashed gloved hands having been shown to be a poor substitute for correct hand hygiene [13]. Anecdotally at the study site, a hospital in London, U.K., an increase in glove use and decrease in correct HHC had been noted by infection control teams as the hospital shifted to more typical pre-pandemic infection control practices.

Behavioural science principles can be applied to create "nudges"–interventions and policies aimed at encouraging individuals to engage in certain behaviours, based on the principle that such behaviours are often influenced in automatic ways by the surrounding context [14]. The process of creating a nudge includes identifying the determinants of the desired behaviour–(e.g. what environments or opportunities will facilitate adoption of the behaviour). Once these are identified, they can be mapped to known behaviour change techniques, such as the use of incentives or training [15]. Behavioural science can improve the effectiveness of interventions by understanding and directly addressing the determinants of the target behaviour, ensuring the intervention is specific to the context, and making it preferable for individuals to adopt or choose to undertake the behaviour itself [16].

The impact of an intervention can also be enhanced by adopting co-design methodology. Co-design is an approach to the creation of an output (be that research, an innovation or intervention) that incorporates the meaningful involvement of end-users [17]. Participants are involved throughout the entire design process, shaping and informing the outputs. It is hoped that by utilising co-design methodology, resulting nudges are more likely to be impactful, as they are co- created by the users they are targeted at.

Behavioural science has been applied to the development of many interventions aimed at improving HHC worldwide. These include the use of olfactory and visual primes, placed at key locations where individuals would be expected to wash their hands, with a variable effect on compliance [18–20]. The type of visual prime is important–with studies showing an image of male eyes to be more effective than female [18], and baby eyes were more effective than either pictures of adult eyes or flowers [19, 21]. Multimodal behavioural nudging strategies

(such as friendly competition, monitoring and feedback, ergonomic placement of alcohol gel, and team training [7, 22, 23]) and educational interventions [24, 25] have also improved HHC.

The rationale for this study was to evaluate current hand hygiene practice at the study site in response to observed non-compliance and ongoing incidences of HCAI. We aimed to explore the reasons for non-compliance with hand hygiene practices, identify the determinants of the target behaviour, and co-design a behavioural-science informed intervention to improve HHC.

Specific objectives were:

1. Explore current hand hygiene practices and identify determinants of the target behaviour through observation.

2. Co-design a behaviourally-informed intervention with infection control and nursing staff, with the objective of increasing hand hygiene compliance

3. Implement and evaluate the effectiveness of a behavioural science informed intervention on hand hygiene compliance.

## Materials and methods

This work was conducted as a quality improvement study using observational measurements of behaviour; and is reported in line with the SQUIRE 2.0 guidelines [26].

This study was approved as a service evaluation and quality improvement project by the clinical governance team at the study site Our data collection process was the same as standard practice for hand hygiene auditing at the study site which are carried routinely at regular intervals. As such, the requirement for individual participant informed consent was waived by the study site. Each participating study site's ward manager consented to the project taking place. All methods within this study were carried out in accordance with the relevant guidelines. Studies involving NHS staff recruited as participants by virtue of their professional role do not require formal ethics approval, in accordance with national guidance provided by the National Health Service Health Research Authority (www.hra-decisiontools.org.uk). All study protocols were approved by the NHS Trust's Governance Team.

### Study setting and target wards

This study was conducted in one large tertiary NHS hospital, one of three hospitals making up an NHS Trust in London. Within this site, three wards were selected to be the focus of our evaluation and intervention. These wards were selected as they demonstrated a range of HHC in previous trust audit processes from low to high compliance. The selected wards also ensured that a range of clinical specialties and acuities were represented (medical, surgical and acute care), and as such as a whole represented a typical inpatient population. It was anticipated this would increase the transferability of our intervention to other similar inpatient wards in the hospital if successful.

The study population included all healthcare staff working on the target wards during the study period, including agency staff, as this reflected the typical staffing at the study site. This was to ensure as many determinants of HHC as possible were captured, with the anticipation that behaviours may differ between different staffing groups. This also allowed a true measure of HHC across the different professional groups working on the ward (doctors, nurses, physiotherapists, phlebotomists and healthcare assistants) to be obtained, both pre- and post-intervention. At the time of this study, no other initiatives to improve HHC were being undertaken on the three target wards.

Within this study, HHC was defined as correctly using either alcohol-based handrubs or washing hands with soap under running water, prior to an interaction with a patient. At the study site, alcohol-based hand rubs were present on every patient's bed, as well as at the entrance to the wards and each patient bay. Hand washing stations were present at each patient bay, typically containing six beds. There was no change in the provision of HHC facilities during the study period.

## Study design

This quality improvement study utilised mixed methodology. A qualitative evaluation of the determinants of HHC behaviours was undertaken through observations, alongside an observed quantitative measure of the proportion of correct HHC before and after the implementation of a co-designed behavioural nudge intervention.

## Understanding determinants of the target behaviour

Determinants of the target behaviour (HHC) were evaluated through the qualitative observation of staff on the three target wards. Eighteen hours of observations of HHC were undertaken over six weeks by two researchers trained in qualitative methodology (CM, KG) (divided equally between the wards). These were focused on two moments for hand hygiene within the WHO five moments for hand hygiene–five points during a typical patient encounter where HHC should take place [5]. The two moments selected were Moment 1 (before touching a patient) and Moment 5 (after touching a patient's surroundings), as previous audit data from the study site had demonstrated the lowest compliance with these two moments. Observations took place at varying times throughout the day to ensure a range of ward activities were captured. Handwritten notes were taken during these observations and analysed thematically by two researchers (CM, KG). 10% of observations were undertaken by both researchers together to ensure inter-rater reliability. Insights relating to the determinants of behaviour were identified and discussed with the wider study team for agreement.

Behaviours identified through the thematic analysis as either barriers or facilitators to HHC were mapped to the COM-B framework [27]. This is a behavioural science framework which allowed the team to evaluate these behaviours in the context of three factors which allow the target behaviour to occur: *capability* (for example, does an individual feel they have the knowledge to conduct a behaviour?), *opportunity* (is the environment conducive to enacting the behaviour?) and *motivation* (is there intrinsic and extrinsic motivation for an individual to conduct the behaviour?).

## Co-design and development of the intervention

Identified determinants of behaviour were used to inform the design and development of the intervention in the co-design process, alongside staff and study team experiences and background literature. A series of workshops were conducted over a four week time frame with the objectives of exploring the barriers to correct hand hygiene and generating ideas for potential behaviour change interventions. These workshops were planned to take 3 hours each, with 8–10 participants invited to attend each one. Participants included the research and design professionals from the study team, the infection control team from the study site and health professionals from the three target wards.

During the first workshop, the determinants of HHC behaviour were reviewed and discussed, with the group achieving consensus on which key barriers and areas to focus the intervention design upon. Co-design participants were then asked to generate as many ideas as possible for interventions that might achieve the objective of improving HHC, using the

COM-B framework as a guide to ensure each idea was focused upon a determinant of the target behaviour. Participants were then asked to rate each idea from 1–5 according to both their perceived impact on HHC and feasibility for creating and deploying the intervention in the healthcare setting. Prior to the second workshop, the study team used these ratings to distil the ideas into a shortlist. This shortlist was discussed at subsequent workshops, with participants discussing each idea in small groups, developing each idea and presenting their top two interventions to the rest of the group. Consideration was also given to the locations on the wards where the behaviours were occurring, to provide context and rationale for the location of the intervention when deployed. All shortlisted ideas were reviewed in the context of the findings of the baseline observations and the WHO moments for compliance, and again mapped to the COM-B framework, to ensure that the final intervention truly reflected the behaviour the study was aiming to change.

### Implementation of the intervention

The final intervention was launched on the three target wards simultaneously, and was planned to run for a total of six weeks.

### Study team

The team implementing this project were a multidisciplinary group of behavioural scientists and qualitative researchers (KG, AB, CM, HB, CJB, FOD, SH) (five of which had a clinical background (KG, AB, CM, FOD, SH)). This team all have prior experience of conducting behaviourally-informed research in the clinical setting. These team members were supported by a senior infection control nurse (TG), senior nursing colleague (CLV), a designer (AG), and members of the medical director's office (KC, SM, RA).

### Outcome measures and data collection

Quantitative measures of the proportion of HHC with Moments 1 and 5 were obtained through observations by two researchers (CM, KG). Eighteen hours of observation were undertaken pre-intervention (aiming for six hours per ward) to provide a baseline measurement of HHC. Following implementation, 18 hours of observation of Moment 1 and Moment 5 were conducted over the six-week intervention period by the same researchers, again divided equally across the three wards and at various points within the working day. These two researchers were clinically trained and understood the clinical indications for HHC. Each observation period was planned to last for one hour, with three hours conducted each week on separate days. Staff were observed as they conducted their routine tasks, recording any behaviour that required Moment 1 or Moment 5 HHC to be conducted, alongside an additional record of whether this moment was compliant. These were recorded with handwritten notes and transferred to a Microsoft Excel spreadsheet. All staff groups working on the ward were included in these observations.

Observers moved around the ward to ensure all staff working were observed during the observation period, and that the focus was not on one member of staff or one clinical team.

### Data analysis

Quantitative data on HHC across the three target wards was non-parametric. In order to evaluate any changes in HHC as a result of the intervention, inferential statistical analysis was conducted in the form of a comparison of the proportions of correct HHC observed. Whilst the three wards were the same location pre- and post-intervention, the staff members present on

each ward were different during each observation period (reflecting shift patterns, agency staff and visiting clinical teams). As such, these data were treated as unpaired, and a Chi-squared statistical test applied.

This study was designed as a quality improvement project, intended to evaluate the impact of an intervention across the entire study population (all observed participants across the three target wards). An indicative power calculation was conducted to detect a difference of 10% in HHC pre- and post-intervention. Accepting a two-sided p-value of <0.05 as statistically significant and 80% power, the recommended sample size in each group was 157. However, as the team wanted to capture the effect of the intervention over the full 6-week period, observations were carried out for the full 18 hours, even once this target sample size had been achieved. Sub-group analyses evaluating the effect of the intervention for each moment and each ward were conducted to provide further information about the intervention's effect, however the study was not explicitly powered to detect a difference in these sub-groups.

## Results

### Identifying determinants of the target behaviour

Eighteen hours of qualitative observation were undertaken. One ward (Ward 3) joined the study later in the pre-observation period due to clinical constraints and was observed for three hours, with the other two wards observed for 7.5 hours each. Observed determinants of behaviour were identified and categorised into facilitators and barriers to HHC and mapped to the COM-B framework (Table 1).

Three key touchpoints were identified as occasions where HHC was most likely to be missed:

- Using the "computer on wheels" to complete tasks during patient interactions, and subsequently forgetting to complete HHC on returning to the patient.

**Table 1. Determinants of behaviour identified through qualitative observation relating to hand hygiene compliance (HHC), and their relationship to the COM-B framework (Capability, Opportunity, Motivation, Behaviour).**

| Category of determinant | Determinant Identified | Relationship to COM-B Framework |
|---|---|---|
| Barriers | Forgetting to engage in HHC | Capability |
| | Interruptions during a task/process | Opportunity |
| | The presence / availability of gloves | Opportunity |
| | Gloves seen as an alternative to HHC | Motivation, Opportunity |
| | Confusion between Moment 5 ending and Moment 1 beginning | Capability |
| | Non-clinical encounter anticipated, but patient contact taking place | Capability, Opportunity |
| | Lack of role-modelling by senior staff | Motivation |
| | Negative ward culture / team culture around HHC | Motivation |
| | Limited bandwidth (multiple competing factors for attention) | Opportunity |
| | Different baseline "norms" for best practice following COVID-19 | Capability |
| Facilitators | Positive ward culture / team culture around HHC | Motivation |
| | Leadership and accountability | Motivation |
| | Presence of role-modelling by senior staff | Motivation |
| | Training and understanding | Capability |

- Use of gloves as an alternative to hand hygiene compliance and not washing/gelling hands before or after patient contact.

- When approaching the patient's bedside–HHC not completed before touching the patient.

Our observations highlighted that these were particularly relevant for compliance with Moment 1. These three touchpoints formed the basis of the discussion in the co-design workshops on where to locate our potential interventions.

## Co-design of the intervention

The first co-design workshop included discussion with frontline and infection control staff regarding the known barriers and facilitators to HHC, to evaluate agreement with the determinants of behaviour identified in our qualitative observations.

Following the discussion of barriers / facilitators and target behaviours, participants were asked to generate as many ideas as possible for potential interventions, leading to the creation of over 150 unique ideas. The group also deemed that Moment 1 should be the focus of our intervention, given the historical low compliance with this moment in previous trust audits, and that our baseline observations highlighted that Moment 1 was more likely to be the point at which HHC failed.

These 150 ideas were rated by the co-design team according to perceived impact and feasibility, leading to the creation of 12 shortlisted ideas. These were mapped to the COM-B framework, to evaluate the key areas of behaviour each idea might target: capability, opportunity, and motivation. Shortlisted ideas and their corresponding COM-B element can be seen in Table 2.

Each shortlisted idea was reviewed at a second workshop, with small group activities to consider what each intervention might look like. For example, a physical prompt could include messaging on curtains, incentives could take the form of pleasantly scented hand gel. These ideas were again reviewed for potential impact and feasibility, and the shortlist refined accordingly. The team also considered other factors, such as the benefits of patient empowerment, changes in patient and staff perceptions of what is "acceptable" post covid (particularly with respect to glove use), and initiatives at the study site to be more sustainable.

**Table 2. Ideas for interventions that might improve hand hygiene compliance that were shortlisted during the co-design process, mapped to elements of the COM-B behaviour change framework.**

| Shortlisted Intervention Idea aimed at improving hand hygiene compliance (HHC) | Factor identified within the COM-B Framework that would allow the target behaviour to occur |
|---|---|
| Physical prompt in ward environment | Motivation, Capability |
| Addressing the use of gloves as an alternative | Opportunity, Capability |
| Use of lighting / motion sensors to prompt HHC | Motivation, Opportunity |
| Incentives to use hand gel | Motivation, Opportunity |
| Patient empowerment / education to prompt HHC by healthcare professional | Capability, Motivation |
| Creating a ward culture that promotes HHC | Motivation, Opportunity, Capability |
| Use of social media to promote HHC | Motivation |
| Modifying ward ergonomics (e.g., sinks, hand gel location) | Opportunity |
| Estates support to ensure HHC possible | Opportunity |
| Use of glo-boxes to highlight need for HHC | Motivation, Capability |
| Highlighting the consequences of poor HHC | Motivation |
| Highlighting environmental consequences of unnecessary glove use | Motivation |

A final shortlist was created by considering the previous impact and feasibility scoring (with ideas scoring highly across both parameters taken forwards), review with infection control and nursing staff, and ongoing review with behavioural scientists. Observational data were mapped (particularly focusing upon the three key touchpoints where Moment 1 was appearing to fail) to the intervention ideas, considering the best ways to motivate individuals to complete Moment 1. The final shortlist included three ideas—"physical prompt", "addressing the use of gloves" and "optimising hand gel use".

Building on this process of co-design, the team decided to focus on "physical prompts" as the intervention to be tested, in the form of a visual prime. This was selected based upon its perceived likely impact and the feasibility of implementing such an intervention. The intervention would take the form of stickers that were placed in the three previously identified key touchpoints where Moment 1 compliance failed. Other strong themes from the co-design process were incorporated, including patient empowerment, ensuring the patient's voice was represented, addressing the use of gloves, and considering the environmental impact of unnecessary glove use.

Three sticker designs were created, each one targeting one of the three key touchpoints identified earlier in the project (the computer on wheels, the patient's bed and the use of gloves). Each design was to be placed at the point where these touchpoints occurred on the ward–e.g. for use of gloves, the stickers were to be placed on the glove dispenser boxes. These were printed to align with infection control requirements and were easy to remove, replace or reposition. Each design contained two components–one identifying the behaviour where HHC is needed (positioned on the left), and the second with a "call to action" to complete hand hygiene practices (positioned on the right) (Fig 1).

## Implementation of the intervention

The three sticker sets that comprised the final intervention were successfully launched on all three wards, with no unintended consequences or harms observed. The three locations (computer on wheels, patient's beds and glove boxes) for the visual primes were present across all three target wards. 20 visual primes were placed on computer on wheels across the three wards, 89 on patient's beds, and 28 on the glove boxes.

## Evaluation of hand hygiene practices pre and post intervention

Staff conducting HHC were observed pre-and post-intervention on all three wards. All staff working on the wards were represented in our sample, including doctors, nursing staff, physiotherapists, phlebotomists and healthcare assistants.

Pre-intervention, across the entire study population (all three wards and both moments), 608 moments were observed, of which 256 were compliant (42%). Across all three wards, Moment 1 had lower compliance overall (31%), when compared to Moment 5 (49%). The three wards themselves had compliance of 40%, 41% and 48% respectively (Table 3).

Post-intervention launch, a further 18 hours of observation were conducted evenly across the three wards over the subsequent 6 weeks by the same two researchers (KG, CM). Table 3 shows overall compliance pre-and post-intervention, with subgroup analyses according to both ward and moment.

Post-intervention, across all three wards and both Moments 1 and 5, 677 moments were observed. Of these, 355 were compliant, giving an overall compliance of 53%. There was an absolute increase in HHC of 11% post intervention which was statistically significant ($X^2$ (1, N = 1,285) = 13.711, p = <0.001).

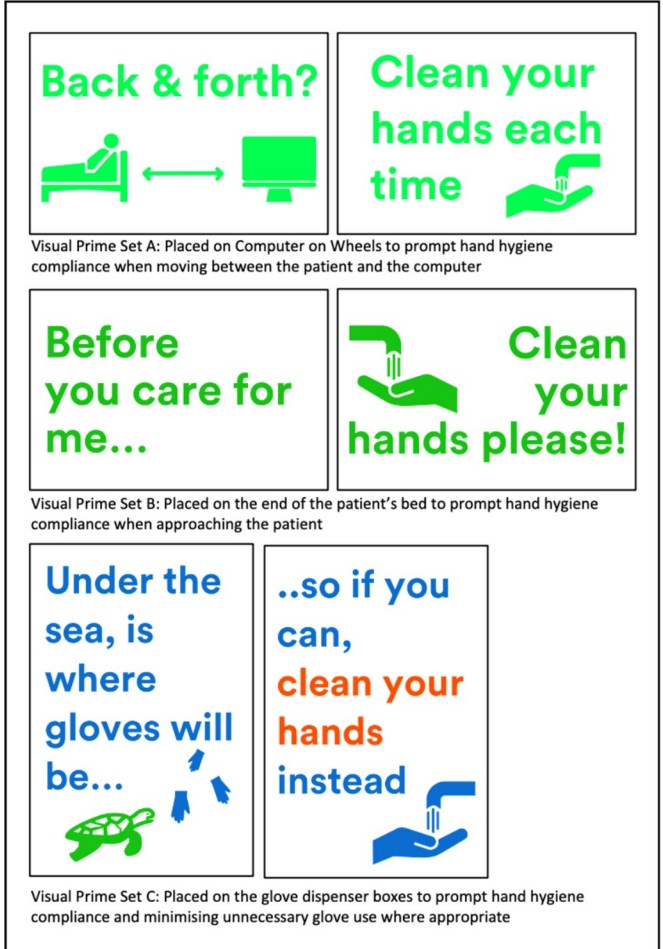

**Fig 1. The three visual prime interventions co-designed to improve hand hygiene compliance.**

Looking at HHC for both moments 1 and 5 individually, there was a statistically significant absolute increase in HHC with Moment 1 of 15% (from 31% pre intervention to 46% post intervention ($X^2$ (1, N = 667) = 17.091, p = <0.001). Additionally, an absolute rise in the

**Table 3. Observed hand hygiene compliance (HHC) pre and post intervention.** Change in compliance following the intervention shown in %, and as a comparison of proportions using Chi-squared statistical analysis. *denotes statistical significance.

| Area of observation | | Compliance with HHC pre-intervention (moments observed to be compliant / total number of moments observed (%)) | Compliance with HHC post-intervention (moments observed to be compliant / total number of moments observed (%)) | Increase in compliance with hand hygiene compliance following intervention launch (%), $X^2$(degrees of freedom, N- sample size) = chi-square value, p = p value) |
|---|---|---|---|---|
| All wards / moments | | 256/608 (42%) | 355/677 (53%) | 11%, $X^2$ (1, N = 1,285) = 13.711, p = <0.001* |
| Ward | Ward 1 | 91/227 (40%) | 91/217 (42%) | 2%, $X^2$ (1, N = 444) = 0.157, p = 0.692 |
| | Ward 2 | 106/259 (41%) | 150/261 (57.5%) | 16.5%, $X^2$ (1, N = 520) = 14.237, p = <0.001* |
| | Ward 3 | 59/122 (48%) | 114/199 (57%) | 9%, $X^2$ (1, N = 321) = 2.425, p = 0.119 |
| WHO Moment for hand hygiene observed | Moment 1 | 87/284 (31%) | 178/383 (46%) | 15%, $X^2$ (1, N = 667) = 17.091, p = <0.001* |
| | Moment 5 | 176/357 (49%) | 177/295 (60%) | 11%, $X^2$ (1, N = 652) = 7.449, p = 0.06 |

proportion of correct HHC with Moment 5 of 11% was seen, but this did not reach statistical significance ($X^2$ (1, N = 652) = 7.449, p = 0.06).

When the change in HHC was evaluated per target ward, there were differences in the size of the increase of the proportions of compliant HH practice. All three wards demonstrated an increase, with one ward demonstrating a statistically significant change (Ward 2, absolute increase of 16.5% ($X^2$ (1, N = 520) = 14.237, p = <0.001). All results can be seen in Table 3.

## Discussion

### Key findings

This study demonstrates how behaviourally informed and co-designed interventions can be deployed in the clinical environment to successfully increase the rate of hand hygiene compliance. The study highlights the importance of thoroughly understanding the target behaviour (in this case HHC) and identifying the reasons why the behaviour might not take place. This process ensures that the intervention can focus upon overcoming these barriers, increasing its likelihood of success.

We believe a key element in the success of our intervention was the co-design process used to design the visual primes. Building effective relationships between researchers and designers in the study team and staff at the NHS study site (including frontline nurses and infection control staff) meant that the intervention was created by the individuals who it would directly affect, and incorporated input from those who truly understood the clinical environment and associated workflows. It also meant we targeted key areas where HHC compliance was failing–by triangulating our qualitative observations with this frontline knowledge. Engaging nursing staff from the target wards engaged in the co-design process was anecdotally perceived by the study team to assist with implementation of the intervention, and likely contributed to its effectiveness.

### Study findings in the context of existing research

Results from our study correlate with previous work implementing behaviourally-informed visual primes–demonstrating that this is an effective strategy in motivating and reminding healthcare staff to complete correct HHC [18, 28, 29]. We build on existing literature exploring the benefits of co-design for the design of hand hygiene compliance [30, 31] by not simply generating the idea for the intervention, but successfully implementing the intervention in collaboration with the co-design team.

### Strengths

There are several strengths of this study which may explain why this intervention was successful. Firstly, through the co-design process the team were able to truly appreciate the clinical setting for the intervention, incorporating the knowledge that staff are under extreme time pressures and that the ward environment is often very hectic with a lot of duties to be carried out in a small timeframe into the intervention design, ensuring it was eye-catching and easy to interpret quickly. This process also increased engagement between the study team and clinical staff, improving the set up and delivery of the intervention on the target wards.

Utilising qualitative observations allowed the team to thoroughly interrogate why HHC might be lower than desired and the determinants of this behaviour–leading to the specific targeting of these during the intervention design. Mapping these insights to the COM-B framework and utilising this to refine our intervention ensured the design helped to overcome key barriers. We believe this process (in combination with the co-design) enhanced our

intervention's effectiveness, as we understood the target behaviour in the context, and applied behavioural science principles to nudge individuals into completing HHC.

Additionally, the observations allowed us to identify where on the ward these behaviours were occurring. By placing our interventions directly in these areas in the clinical areas (and therefore in the line of sight of staff members exactly at the moment they are required to practice hand hygiene) is likely to have contributed to their success.

## Limitations

There are several limitations within this study. Due to the team's capability and the feasibility for delivering the study, and the fact that this was a safety priority for the study site (hence a requirement to generate results quickly), we did not randomise wards to receive the intervention, and assessed improvement by conducting observations pre and post intervention, rather than comparing against control wards. As such, our conclusions may not be as robust as if generated in a randomised controlled trial. The lack of a control group also limits our ability to evaluate the presence of a possible Hawthorne effect. Further evaluation of the intervention with a control group would be of benefit to understand the potential presence of this effect.

The use of observations as a measurement tool may also have introduced bias–despite best efforts by the researchers, it is possible that there is a Hawthorne effect [32] contributing to the increase in HHC–healthcare staff may have recognised they were being observed as part of a HHC project.

The observation period only extended for 6 weeks following the introduction of the intervention. As such, any sustained impact beyond this was not evaluated.

This study was conducted in three wards in a large tertiary hospital. As such our findings may not be generalisable to other clinical settings without a prior exploration of the specific barriers to HHC present. However, given the careful selection of the three target wards (ensuring they were representative of a range of typical adult inpatient environments), we anticipate our intervention will be transferable to other similar clinical environments.

Whist the overall study sample (across all three wards), and the subgroups of moment 1/ moment 5 were adequately powered to detect a 10% change in HHC, the subgroup analysis of the effect of our intervention per ward itself was not adequately powered. This analysis was not an explicit goal of the study, however this subgroup analysis must be interpreted with caution– there is a signal that there is a difference between wards, but further powered studies should be undertaken to provide a robust conclusion and understand why this may have occurred.

Finally, our study tested three different designs all within one intervention–targeting three different behaviours where moment 1 had been identified as failing. It may be that one of the three interventions was more impactful than the others. This provides an opportunity for future work–designing a research study in which wards are randomised to receive one of the three interventions, to evaluate whether individual components are as effective as the combined set, and the level of the associated improvement.

## Opportunities for future work

There are several opportunities for future work. It would be interesting to explore whether our intervention is effective at maintaining a rise in hand hygiene compliance and determining the duration of its effectiveness at improving HHC. This could take the form of a longer observational study. It would also be useful to conduct an objective evaluation of our intervention's effectiveness alongside ongoing observation–perhaps by measuring a change in hand gel / soap consumption or glove use.

It would be of benefit to trial our intervention in other healthcare settings (in addition to other adult inpatient wards), such as outpatient departments or paediatric areas.

Our intervention demonstrated a significant but modest improvement in HHC, suggesting that a multi-modal approach to improving HHC should be adopted. Future co-design work with key stakeholders should aim to create a suite of interventions, including visual reminders, educational initiatives and communications campaigns.

It would also be of merit to use this study and the growing body of evidence regarding the effectiveness of behavioural science informed interventions (and understanding of why target behaviours such as HHC are not occurring) to further advocate for the application of behavioural science and co-design methods in the creation of patient safety initiatives.

## Conclusion

The implementation of a co-designed, behaviourally informed visual prime intervention was effective at increasing the level of HHC at our study site. By utilising behavioural science frameworks and qualitative observation to understand the key behaviours associated with poor HHC, the study ensured that the interventions were focused upon areas within the healthcare environment where an improvement in behaviour was most needed. This study highlights that a low cost and easy to implement intervention can be extremely effective, particularly when it is co-designed with individuals who have extensive knowledge of the clinical setting.

## Acknowledgments

We would like to thank the staff on our study wards who supported the implementation of this study. Infrastructure support for this study was provided by Imperial College Institute of Global Health Innovation and Imperial College NHS Healthcare Trust. The views expressed are those of the author(s) and not necessarily those of the NHS.

## Author Contributions

**Conceptualization:** Clare McCrudden, Sarah Huf, Tracey Galletly, Helen Brown, Kristina Clay, Kate Grailey.

**Data curation:** Clare McCrudden, Kate Grailey.

**Formal analysis:** Kate Grailey.

**Methodology:** Alix Brazier, Alice Gregory, Fiona O'Driscoll, Tracey Galletly, Clare Leon-Villapalos, Helen Brown, Kristina Clay, Kate Grailey.

**Project administration:** Clare McCrudden, Kate Grailey.

**Supervision:** Sarah Huf, Shona Maxwell, Raymond Anakwe.

**Writing – original draft:** Calea-Jay Blair, Kate Grailey.

**Writing – review & editing:** Clare McCrudden, Alix Brazier, Sarah Huf, Alice Gregory, Fiona O'Driscoll, Tracey Galletly, Clare Leon-Villapalos, Helen Brown, Kristina Clay, Shona Maxwell, Raymond Anakwe.

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
