## [Decision Letter · Decision Letter 0]

28 Aug 2024

PONE-D-24-28481A helping hand: applying behavioural science and co-design methodology to improve hand hygiene compliance in the hospital setting.PLOS ONE

Dear Dr. Grailey,

Thank you for submitting your manuscript to PLOS ONE. After careful consideration, we feel that it has merit but does not fully meet PLOS ONE’s publication criteria as it currently stands. Therefore, we invite you to submit a revised version of the manuscript that addresses the points raised during the review process.

We look forward to receiving your revised manuscript.

Kind regards,

Juraj Ivanyi

Academic Editor

PLOS ONE

Reviewers' comments:

Reviewer's Responses to Questions

**Comments to the Author**

1. Is the manuscript technically sound, and do the data support the conclusions?

Reviewer #1: Partly

Reviewer #2: Yes

2. Has the statistical analysis been performed appropriately and rigorously? 

Reviewer #1: Yes

Reviewer #2: Yes

3. Have the authors made all data underlying the findings in their manuscript fully available?

Reviewer #1: Yes

Reviewer #2: Yes

4. Is the manuscript presented in an intelligible fashion and written in standard English?

Reviewer #1: Yes

Reviewer #2: Yes

5. Review Comments to the Author

Reviewer #1: This study addresses a relevant topic; correct hand hygiene is a safety priority in all healthcare environments. The report is well structured, the objective is clear, and the study’s context is well described.

The results show that the intervention, a motivational nudge in the form of visual primes, has the potential to obtain a modest improvement in hand hygiene compliance. However:

1) Authors must include a control to assess how much of the eleven percent improvement is attributable to a possible Hawthorne effect.

2) Authors must determine the duration of the stickers’ nudging effect. It’s possible that visual reminders eventually become part of the “scenery” and are no longer noticeable to the target population.

3) The modest improvement obtained must prompt the authors to explore multimodal behavioral nudging, educational interventions and an enforcing program to achieve sustained compliance. The desirable outcome is that all end-users contribute to the ward’s safety culture.

The results show the magnitude of the challenge, close to half of the patient encounters (moments one and five, in the three wards, before or after the intervention) were lacking in hand hygiene compliance. This is a daily occurrence at a hospital in a country where healthcare professionals have access to the highest standards of education and training, as well as to the needed infrastructure and supplies.

In the study’s design, it’s not explicit what hand hygiene protocols were expected from participants and recorded by the observers. Apparently, authors aimed to optimize hand hygiene with the use of alcohol-based hand gel. Indeed, the visual primes presented (“clean your hands each time”, “clean your hands please!” and “.. so if you can, clean your hands instead”) suggest the use of alcohol-based handrubs.

Alcohol-based handrubs provide an alternative for hand hygiene. This method is accepted only if hands are not visibly soiled with dirt, and no contact with bodily fluids has occurred. Authors must describe how the handrubs were made available and within reach, either at bedside or for individual carriage.

In this report, hand washing with soap under running water, followed by proper drying of hands, is not explicitly excluded. If handwashing under running water was included in the study, this form of compliance required the availability of handwashing stations in each ward. Authors should explain whether differences in the numbers of beds and sinks, as well as the distance to the nearest handwashing station had an impact on compliance, before and after the intervention.

The acronym NHS is not spelled out

Under Background, the abbreviation “HCC” appears 6 times before being “introduced” in line 106.

Typo “_” is present in lines 155, 330

Reviewer #2: I found the reviewed work to be very good. The entire methodology of the work seems very appropriate and the results very interesting. In addition, the specific topic of hand hygiene is a topic that, although it has been widely covered in many studies, still has many improvement objectives to achieve.

6. PLOS authors have the option to publish the peer review history of their article (what does this mean?). If published, this will include your full peer review and any attached files.

Reviewer #1: No

Reviewer #2: No

---

## [Author Response · Author response to Decision Letter 0]

4 Sep 2024

Response to Reviewers

A helping hand: applying behavioural science and co-design methodology to improve hand hygiene compliance in the hospital setting. 

We would like to thank the Editors and Reviewers for their thorough review of our manuscript. We have made edits to the manuscript to address these points, without significant alterations to the previous analyses or conclusions drawn. Please find our point-by-point response in line below.

Reviewers' comments:

Reviewer #1: This study addresses a relevant topic; correct hand hygiene is a safety priority in all healthcare environments. The report is well structured, the objective is clear, and the study’s context is well described.

Thank you for your review. 

The results show that the intervention, a motivational nudge in the form of visual primes, has the potential to obtain a modest improvement in hand hygiene compliance. However:

1) Authors must include a control to assess how much of the eleven percent improvement is attributable to a possible Hawthorne effect.

Thank you for highlighting this important point. We acknowledge that there is the possibility of a Hawthorne effect and that this could be responsible for some of the improvement. We did not conduct this study with a control group, and instead used a pre- post study design. We acknowledge this is a limitation of the work and have now included this within the limitations section of the manuscript. This demonstrates an opportunity for future evaluations of our intervention, conducting a more formalised trial with a control group (Lines 418-421). 

2) Authors must determine the duration of the stickers’ nudging effect. It’s possible that visual reminders eventually become part of the “scenery” and are no longer noticeable to the target population.

Thank you. Visual reminders becoming a part of the “scenery” is a real challenge. During the 6 weeks of our intervention period we did not notice a change in the effect when broken down according to each week – there was no decrease in hand hygiene compliance during this time. Further work to determine the duration of the effect would require a longer observational study – we have added this to the discussion (Lines 448-449)

3) The modest improvement obtained must prompt the authors to explore multimodal behavioral nudging, educational interventions and an enforcing program to achieve sustained compliance. The desirable outcome is that all end-users contribute to the ward’s safety culture.

The results show the magnitude of the challenge, close to half of the patient encounters (moments one and five, in the three wards, before or after the intervention) were lacking in hand hygiene compliance. This is a daily occurrence at a hospital in a country where healthcare professionals have access to the highest standards of education and training, as well as to the needed infrastructure and supplies.

We agree with this wholeheartedly, and have expanded upon the need for multi-modal interventions created through a co-design process in the discussion (Lines 455-458). 

In the study’s design, it’s not explicit what hand hygiene protocols were expected from participants and recorded by the observers. Apparently, authors aimed to optimize hand hygiene with the use of alcohol-based hand gel. Indeed, the visual primes presented (“clean your hands each time”, “clean your hands please!” and “.. so if you can, clean your hands instead”) suggest the use of alcohol-based handrubs.

Alcohol-based handrubs provide an alternative for hand hygiene. This method is accepted only if hands are not visibly soiled with dirt, and no contact with bodily fluids has occurred. Authors must describe how the handrubs were made available and within reach, either at bedside or for individual carriage.

In this report, hand washing with soap under running water, followed by proper drying of hands, is not explicitly excluded. If handwashing under running water was included in the study, this form of compliance required the availability of handwashing stations in each ward. Authors should explain whether differences in the numbers of beds and sinks, as well as the distance to the nearest handwashing station had an impact on compliance, before and after the intervention.

Thank you for highlighting this, and we apologise that there was not enough clarity on this in the original manuscript. Our aim was to optimise hand hygiene compliance through either the use of alcohol gel or hand washing with soap under running water. This is why we used the language “clean” in our visual primes, as it was felt by our co-design team of experts and frontline staff to encompass both methods. We did not deliberately change the availability of alcohol hand rubs or hand washing stations as part of our intervention, to allow us to solely evaluate the impact of our visual primes. 

More detail on the inclusion of both HHC techniques and the presence of alcohol hand rubs / hand washing stations throughout the study period has been included within the methods section (Lines 150-155). 

The acronym NHS is not spelled out

Thank you for highlighting this – this is now corrected. 

Under Background, the abbreviation “HCC” appears 6 times before being “introduced” in line 106.

Thank you for highlighting this – this is now corrected. 

Typo “_” is present in lines 155, 330

Thank you for highlighting this – this is now corrected. 

Reviewer #2: I found the reviewed work to be very good. The entire methodology of the work seems very appropriate and the results very interesting. In addition, the specific topic of hand hygiene is a topic that, although it has been widely covered in many studies, still has many improvement objectives to achieve.

Thank you for your review.

---

## [Editor Report · Decision Letter 1]

6 Sep 2024

A helping hand: applying behavioural science and co-design methodology to improve hand hygiene compliance in the hospital setting.

PONE-D-24-28481R1

Dear Dr. Grailey,

We’re pleased to inform you that your manuscript has been judged scientifically suitable for publication and will be formally accepted for publication once it meets all outstanding technical requirements.

Kind regards,

Juraj Ivanyi

Academic Editor

PLOS ONE
---

## [Editor Report · Acceptance letter]

22 Sep 2024

PONE-D-24-28481R1 

PLOS ONE

Dear Dr. Grailey, 

I'm pleased to inform you that your manuscript has been deemed suitable for publication in PLOS ONE. Congratulations! Your manuscript is now being handed over to our production team.

Kind regards, 

on behalf of

Dr. Juraj Ivanyi 

Academic Editor

PLOS ONE